# Effectiveness of Strategies for Nutritional Therapy for Patients with Type 2 Diabetes and/or Hypertension in Primary Care: A Systematic Review and Meta-Analysis

**DOI:** 10.3390/ijerph19074243

**Published:** 2022-04-02

**Authors:** Julia Simões Corrêa Galendi, Renata Giacomini Occhiuto Ferreira Leite, Luísa Rocco Banzato, Vania dos Santos Nunes-Nogueira

**Affiliations:** 1Institute of Health Economics and Clinical Epidemiology, Faculty of Medicine and University Hospital of Cologne, University of Cologne, 50924 Cologne, Germany; julia.simoes-correa-galendi@uk-koeln.de; 2Department of Internal Medicine, Botucatu Medical School, São Paulo State University (UNESP), 18618687 Botucatu, Brazil; renataleite578@gmail.com (R.G.O.F.L.); lubanzato@gmail.com (L.R.B.)

**Keywords:** chronic disease, health services research, type 2 diabetes mellitus, hypertensions, primary care, nutrition therapy

## Abstract

A central aspect to the management of type 2 Diabetes Mellitus (T2DM) and hypertension is promoting a healthy lifestyle, and nutritional therapy (NT) can support patients achieving glycemic control and blood pressure targets. This systematic review aimed to evaluate the effectiveness of NT in the management of patients with T2DM and/or hypertension in primary care. Primary outcomes were HbA1c, systolic blood pressure (SBP) and diastolic blood pressure (DBP). Thirty-nine studies were included, thirty on T2DM and nine on hypertension. With a moderate quality of evidence, educational/counseling programs and food replacement programs in primary care likely reduce HbA1c on patients with T2DM (mean difference (MD): −0.37, 95% CI: −0.57 to −0.17, 7437 patients, 27 studies; MD: −0.54, 95% CI: −0.75 to −0.32, 440 patients, 2 studies, respectively). Mediterranean diet for T2DM was accessed by one study, and no difference between the groups was found. Educational and counseling programs likely reduce DBP in patients with hypertension (MD: −1.79, 95% CI: −3.46, −0.12, 2840 patients, 9 studies, moderate quality of the evidence), but the effect in SBP was unclear due to risk of bias and imprecision. Nutritional therapy strategies (i.e., educational/counseling programs and food replacement programs) in primary care improved HbA1c in patients with T2DM and DBP in individuals with hypertension.

## 1. Introduction

Hypertension and diabetes mellitus (DM) are leading causes of cardiovascular disease and premature death. The total number of patients with diabetes mellitus has quadrupled in the past three decades, and it now affects approximately 1 in 11 adults worldwide [1], while the prevalence of hypertension is estimated at 31.1% among adults [2]. Besides, 11.3% of global deaths in 2019 among adults between 20 and 79 years-old were attributed to diabetes [3], and 14% of deaths globally to hypertension, according to data from 2015 [4].

In addition to the high mortality, the morbidity associated with hypertension and diabetes represent a significant economic burden for patients, caretakers, and health systems. The estimated global cost of DM in 2015 was USD 1.31 trillion or 1.8% of global gross domestic product, and 34% of this was attributed to indirect costs such as loss of productivity [5]. Moreover, hypertension and DM are the health conditions with the highest absolute increase in annual US healthcare expenditure over the past three decades [6].

Despite the increasing investments on chronic disease management, an expressive number of patients do not reach treatment targets. A multicenter, cross-sectional, questionnaire-based study conducted in 9 Latin American countries showed that 56.8% of patients with type 2 DM (T2DM) had poor glycemic control (i.e., HbA1c ≥ 7%) [7]. The lowest treatment success was identified in Peru, where only 7.5% achieved metabolic and blood pressure levels as recommended by the American Diabetes Association (ADA) [8]. Likewise, only 13.8% of adults with hypertension in 2010 had their BP controlled worldwide [9]. Although the management of hypertension and T2DM are well stablished, there is a gap between knowledge and attitude that hinders the implementation of successful management strategies [10]. Hence, the coordination of care and patient self-management are the two utmost important aspects that can be promoted in primary care [11]. 

A central aspect to the management of T2DM and hypertension is providing healthy lifestyle education. Nutrition therapy (NT) consists of education and support to help patients adopt healthy eating pattern, what plays a fundamental role in the management of T2DM and hypertension, and its complications [12,13,14]. The ADA recommends that NT should generally promote dietary quality and energy restriction and combine patient preferences and metabolic needs [12]. With regard to dietary quality, several approaches have been studied, such as the Mediterranean diet, the Dietary Approaches to Stop Hypertension (DASH), a low carbohydrate diet and a vegetarian diet [12]. A network meta-analysis has shown that all dietary approaches are effective to improve glycemic control, but the Mediterranean diet had the more significant effect [15]. The DASH and Mediterranean diets also contribute to BP reduction [16].

Systematic reviews have demonstrated the effectiveness of NT and lifestyle education on the management of T2DM [17,18,19,20] and hypertension [21,22]. These systematic reviews either restricted the inclusion criteria to group-based educational programs [20]; or did not consider antihypertensive medications as part of the standard care [22]; or were conducted on specific populations, such as young adults [21], obese patients [17], patients at risk for T2DM [19]. However, none of them exclusively focused on the primary care setting. Hence, the objective of this systematic review was to evaluate the effectiveness of NT programs delivered exclusively in the primary care setting in the management of adult patients with T2DM and/or hypertension. 

## 2. Materials and Methods

This systematic review was conducted following the Cochrane collaboration handbook [23], and is reported according to the preferred reporting items for systematic reviews and meta-analyses (PRISMA) Statement [24]. The protocol was published elsewhere [25], and had been previously registered with the international Prospective register of Systematic Reviews (PROSPERO) under registration number CRD42018118117.

### 2.1. Inclusion and Exclusion Criteria 

We sought to include randomized controlled trials that met the following inclusion criteria (PICO):

Participants (P): adult patients (i.e., aged ≥ 18) diagnosed with T2DM and or hypertension. The diagnosis of T2DM should have been established according to the ADA criteria (i.e., fasting glycaemia > 200 mg/dL associated with classic T2Dm symptoms; glycaemia 2 h after overload with 75 g of glucose ≥ 200 mg/dL; HbA1c ≥ 6.5%) [12]. Hypertension should be characterized by persistent systolic blood pressure (SBP) ≥ 140 mmHg and/or diastolic blood pressure (DBP) ≥ 90 mmHg [26].

Types of intervention (I): Nutritional therapy (NT) programs delivered in a primary care setting which focused on stimulating healthy nutrition for a minimum duration of four months. We considered the nutritional strategies that were provided in addition to, and not instead of, the regular pharmacological treatment to T2DM and/or hypertension. We included: (i) educational or counseling programs addressing nutritional recommendations to reduce calories and dietary fat and lifestyle healthy behaviors, (ii) food replacement program followed by stepped reintroduction of meals; (iii) Mediterranean diet, (iv) DASH diet, (v) low carbohydrate diet, (vi) vegetarian diet, (vii) low glycemic index diet, (viii) high protein diet. The educational and counseling interventions could be delivered by any health care professional, such as dieticians, physical educators, nurses, psychologists, health educators, physicians, and peer-supporters (i.e., trained in some way in the context of the intervention, although having no formal professional or paraprofessional certificated or degree tertiary education).

Comparison (C): Conventional treatment for T2DM and hypertension, which consisted of pharmacological treatment and general healthy lifestyle advice. An episodic consultation with a dietician, nurse, physical trainer, or educator in diabetes for general healthy lifestyle advice or general nutritional orientation were considered conventional treatment if the patients were not provided subsequent follow up. 

Outcomes (O): Primary outcomes were glycemic control and blood pressure control, which were measured by HbA1c (%) and SBP/DBP (mmHg), respectively. The secondary outcomes were frequency of cardiovascular events (acute myocardial infarction and stroke), weight loss (measured as change in weight or in BMI) and death. Outcomes were evaluated at 6, 12 and more than 12 months. 

We excluded studies that included patients aged <18 years old, pregnant women, diagnosis of secondary hypertension. Further exclusion criteria were if the intervention was based exclusively on dietary supplements, too low energy diets (less than 600 kcal/day), if the intervention was not delivered in a primary care setting (i.e., studies that recruited patients in the primary setting, but delivered the intervention elsewhere were excluded), and if there was a cointervention not common to both groups. 

### 2.2. Identification of Studies 

General search strategies were developed for the main electronic health databases: Embase (1980–2019), Medline though PubMed (1966–2019), LILACS (through Virtual Health Library, 1982–2019) and CENTRAL (Cochrane Collaboration Controlled Trials Register, 1982–2019). A second search on the previously mentioned databases was conducted on 27 December 2021. Search strategies included the following descriptors and synonyms: T2DM OR hypertension—AND—primary health care OR community health planning—AND—nutritional therapy OR lifestyle—AND randomized controlled trial. In PubMed and Embase a filter for randomized controlled trials were applied. There was no language restriction. The full search strategy is on the Appendix A. 

The databases searched for eligible studies included: Trip database, SCOPUS, Web of Science, Cumulative Index to Nursing and Allied Health Literature (CINAHL), Australasian Medical Index, and Chinese Biomedical Literature Database. Furthermore, we searched for studies on ClinicalTrials.gov, the Brazilian Registry of Clinical Trials (Rebec) and grey literature, through abstracts published in annals and lectures. References of relevant primary or secondary studies were screened to identify additional eligible studies. Endnote software was used to download references and remove duplicates. 

Two reviewers (RGOFL and JSCG) independently performed the initial screening of titles and abstracts using the software Rayyan QCRI [27]. The studies selected for full-text review were subsequently assessed for adequacy to the proposed PICO. In case of disagreement, there was a consensus meeting between the reviewers and the project coordinator (VdSN-N) for a final decision.

### 2.3. Data Extraction and Risk of Bias of Included Studies

Two reviewers (LRB and JSCG) used a standardized form to extract relevant data of the included studies (i.e., study identification, publication date, country, sample size, follow up, type of intervention and control, baseline characteristics of patients, and outcome results). To assess the risk of bias of included studies, the same two reviewers used the revised version of the Cochrane tool (RoB 2), which considers bias arising from five domains: (i) the randomization process; (ii) due to deviations from intended interventions (including blinding of patients and personnel, balanced baseline characteristics, and report of measurement of compliance with the intended intervention); (iii) due to missing outcome data; (iv) in measurement of the outcome; and (v) in selection of the reported result. To ensure consistency between reviewers, a calibration exercise was performed and in the case of disagreement, a consensus meeting with the project coordinator (VdSN-N) was made. For domain (iii), while we considered a loss of follow up of more than 20% as high risk of bias for RCTs, we allowed a loss of follow up of 30% for pragmatic trials. It was also considered high risk if the loss of follow up was unbalanced between the two groups. For domain (iv) it was considered high risk of bias when the data collection on the outcomes of interest was performed based on patient records instead of a standardized data collection. Finally, for domain (v) we also tracked the published protocols and trials registrations, when available. 

### 2.4. Synthesis and Analysis of Data

The unit of analysis was the data published in the included studies. Similar outcomes in at least 2 studies were plotted in random-effect meta-analyses using the Stata Statistical Software 17 (StataCorp LLC, College Station, TX, USA). The random effects model was chosen as the analytic model for the meta-analysis. Continuous data were expressed as means and SD and the differences between means with 95% CIs were used as estimate of intervention effect. The mean adjusted differences (MAD) were preferred when available. For studies that reported median and interquartile range, the estimated mean of the sample and SD were obtained from Hozo et al. [28].

For the cluster randomized trials, we used a formula suggested by the Cochrane handbook to find the trial’s effective sample size, which is its original sample size divided by the “design effect.” The design effect can be calculated by: 

1 + (*M* − 1) × *ICC,* where *M* is the average cluster size and *ICC* is the intracluster correlation coefficient [29].

Inconsistencies between the results of the included studies were ascertained by visual inspection of forest plots (no overlap of CIs around the effect estimates of the individual studies) and by Higgins or I^2^ statistic, in which I^2^ > 50% indicates a moderate probability of heterogeneity, and by χ^2^ tests, where *p* < 0.10 indicates heterogeneity. In the case of statistical heterogeneity, we used meta-regression to explore the causes of the inconsistency. Meta-regression models (with random effects) were adjusted with the MD as a dependent variable; and months of follow-up, type of intervention, risk of bias, if MAD was used and characteristics of the baseline participants as moderator variables (mean age and mean HbA1c). The Knapp–Hartung correction was used to calculate the significance of the meta-regression coefficients.

### 2.5. Quality of Evidence 

The quality of the evidence of the intervention’s effect estimate was assessed according to the Grading of Recommendations Assessment, Development and Evaluation (GRADE) methodological guideline [30]. To assess the domain imprecision the optimal information size (OIS) was calculated using the Stata Statistical Software 17 (StataCorp LLC, College Station, TX, USA). A change of 0.5% for the outcome HbA1c, 4 mmHg for the outcome SBP, and 3 mmHg for the outcome DBP were considered to be clinically significant. For all outcomes, 0.05 alfa error and 0.20 beta were assumed. The quality of the evidence was rated down for imprecision if the OIS criterion was not met [30].

## 3. Results

We identified 6363 references on the first search and 1099 references on the second search, following the removal of duplicates. After reading the title and abstract, 124 references were selected for full reading. The 85 excluded references, and the reasons for exclusion are detailed on Appendix A. The main reasons for exclusion were non-randomized trial (3 studies), patients did not meet eligibility criteria (8 studies), wrong intervention (10 studies) or intervention inferior to 4 months (14 studies), intervention not delivered in primary care setting (36 studies), wrong comparator (7 studies), wrong outcome (3 studies), cointervention not common to both groups (3 studies) or study protocol (1 study). Figure 1 shows the study selection process.

### 3.1. Characteristics of Included Studies 

Thirty-nine studies met our eligibility criteria and were therefore included [31,32,33,34,35,36,37,38,39,40,41,42,43,44,45,46,47,48,49,50,51,52,53,54,55,56,57,58,59,60,61,62,63,64,65,66,67,68,69]. Among the included studies, nine evaluated NT primarily for hypertension [61,62,63,64,65,66,67,68,69], 30 evaluated NT primarily for T2DM [31,32,33,34,35,36,37,38,39,40,41,42,43,44,45,46,47,48,49,50,51,52,53,54,55,56,57,58,59,60]. The characteristics of included studies according to the inclusion criteria are detailed in Appendix A and Appendix B.

Nine studies assessed the effect of NT on patients primarily diagnosed with hypertension [61,62,63,64,65,66,67,68,69]. While most studies excluded patients with T2DM, 2 of these studies allowed for patients with concomitant diagnosis of hypertension and T2DM [65,67]. All of the interventions consisted of counseling or educational programs delivered in groups or individually by practice nurses (six studies) [61,62,64,66,68,69], dieticians (two studies) [63,65] and health educators (one study) [67]. 

Thirty studies included patients primarily diagnosed with T2DM, and in all of the studies a proportion of included patients had concomitant diagnosis of hypertension [31,32,33,34,35,36,37,38,39,40,41,42,43,44,45,46,47,48,49,50,51,52,53,54,55,56,57,58,59,60]. The interventions studies consisted of educational and counseling programs in 27 studies [31,32,33,34,35,36,37,38,39,40,41,42,43,44,45,47,48,49,50,51,52,53,54,55,57,59,60], food substitution program in two studies [46,56], and only one studied assessed the effect of Mediterranean diet [58]. The educational/counseling programs were delivered by multi-professional teams in eight studies [35,36,39,41,47,48,53,54], practice nurse in six studies [37,38,40,42,43,59], dieticians (i.e., nutritionists) in five studies [31,34,44,52,60], pharmacists in two studies [45,50], peer supporters in two studies [49,55] and therapists in one study [51]. In three studies, the intervention was delivered by specialists in diabetes or population health management [32,33,57]. Eight studies were cluster randomized trials [31,33,36,48,49,55,63] and three studies despite the cluster randomization, individual patient data were used for the analysis [32,43,50].

### 3.2. Risk of Bias

Risk of bias of included studies is shown in Figure 2 and Appendix A. For the (ii) domain (i.e., deviation from the intended intervention) there were some concerns for most studies, given the behavioral nature of the educational and counseling interventions and lack of blinding of patients and personnel. Among the studies rated as high risk for the (i) domain (i.e., randomization process), three lacked blinding of allocation [44,45,62], and two had unbalances in baseline characteristics (Höchsmann et al. allocated more diabetic patients to the intervention arm and Tejada Tebayas et al. the intervention group had a lower educational profile) [57,63]. 

For the domain missing outcome data, four studies were rated as high risk of bias due to high and unbalanced dropout rates, suggesting that the loss of follow-up was related to the intervention [38,45,48,54]. 

### 3.3. Meta-Analysis of NT for Hypertension

The meta-analysis showed an association between NT and reduction in DBP, MD: −1.79, 95% CI: −3.46 to −0.12, I^2^: 88.1%, 2840 patients, 9 studies, moderate quality of the evidence, Figure 3A, Table 1). However, meta-analysis did not indicate any clear effect of NT in patients primarily diagnosed with hypertension on SBP (MD: −2.17, 95%CI −6.81, 2.46, 4518 patients, 9 studies, I^2^: 88.8%, low quality of the evidence, Figure 3B, Table 1). After visual inspection of the forest plot, the high heterogeneity in both meta-analyses was attributed to one study that overestimated the effect of the intervention [62]. In a sensitivity analysis without Hacihasanoğlu et al. [62], the intervention effect on DBP remained in favor of the intervention, and the MD in SBP still crossed the line of null effect (Appendix A). 

Subgroup analyses performed according to length of follow-up and sensitivity analyses separating studies that presented the results as MAD or post-intervention values, and according to risk of bias did not change the effect on blood pressure (Appendix A, respectively).

### 3.4. Meta-Analysis of NT for T2DM

Meta-analysis of studies assessing the effect of counseling and educational programs on HBA1c favored the intervention (MD: −0.37, 95% CI: −0.57, −0.17, I^2^: 85.8%, 7437 patients, 27 studies, moderate quality of the evidence, Figure 4A). We investigated heterogeneity using meta-regression. The joint and individual tests for categorical and no categorical covariates gave us a *p*-value higher than 0.05, indicating no association between these covariates and the size of the treatment effect (Appendix A). However, through visual inspection of the forest plot, we asserted that two studies overestimated the effect of the intervention [43,45]. A sensitivity meta-analysis excluding these two studies resolved the heterogeneity and still favored the intervention (MD: −0.25, 95% CI: −0.35, −0.16, I^2^: 24%, 7094 patients, 25 studies, Appendix A). 

Subgroup analyses performed according to length of follow-up and sensitivity analyses separating studies that presented the results as MAD or post-intervention values, and according to risk of bias did not change the effect on HbA1c (Appendix A, respectively).

Meta-analysis of studies assessing the effect of food replacement on HBA1c favored the intervention (MD: −0.54, 95% CI: −0.75, −0.32, I^2^: 0%, 440 patients, 2 studies, Figure 4B, moderate quality of the evidence, Table 1). 

#### Secondary Outcomes

Among studies assessing educational and counseling NT for T2DM, the meta-analyses of DBP and BMI did not show a statistical difference between groups (MD: −0.84, 95% CI: −1.85, 0.16, I^2^: 44.8%, 5508 patients, 14 studies, Appendix A; MD: 0.07 95%CI: −0.19, 0.05, I^2^: 24.5%, 2378 patients, 11 studies, Appendix A, respectively). For SBP, a trend of no effect was identified (MD: −2.56, 95% CI: −5.5 to 0.38, I^2^: 83.6%, 5508 patients, 15 studies, Appendix A), but in a sensitivity analysis without Javaid et al. [45] (a study which overestimated the intervention), the heterogeneity was resolved and the reduction in SBP showed to be significant (MD: −1.68, 95% CI: −2.98, −0.38, I^2^: 0%, 5264 patients, 14 studies, Appendix A). 

### 3.5. Publication Bias

Publication bias was investigated for HbA1c outcome. Since asymmetries in the funnel plot was not observed, the Egger test was performed (*p*: 0.988, Appendix A), indicating no publication bias. 

### 3.6. Studies Not Included in the Meta-Analysis

Only 1 study assessed the effect of Mediterranean diet on T2DM, and therefore it was not possible to perform a meta-analysis [58]. Toobert et al. investigated the effect of a nutritional therapy based on Mediterranean diet on 280 patients aged ≥30 and <75 and with Latino ethnicity and the primary outcomes were self-efficacy and behaviour change and HbA1c as secondary outcome. Patients in the intervention groups presented mean HbA1c of 8.4 (standard error (SE): 0.003), which dropped to 7.8 (SE: 0.03) after 6 months, but it was not maintained (i.e., mean HbA1c at 12 and 24 months were both 8.4, SE: 0.03).

The secondary outcomes mortality and frequency of cardiovascular events were not evaluated by the primary studies included in this review, and therefore, were not assessed. 

## 4. Discussion

In this systematic review, we identified three NT strategies evaluated in primary care: counseling and educational programs, food replacement programs and Mediterranean diet. Only one study on Mediterranean diet in primary care was identified [58], and it was not included in a meta-analysis. The included studies reported the metabolic targets as main outcomes, such as glycemic control, blood pressure reduction and BMI. However, no study reported the outcomes mortality or frequency of cardiovascular events.

According to the meta-analyses, NT in primary care likely reduces HbA1c in patients with T2DM. With moderate quality of the evidence, counseling and educational programs were associated with a mean change of −0.37 in HbA1c and food replacement programs with a mean change in HbA1c of −0.54, both when compared to usual care. In addition, counseling and educational programs slightly reduce DBP (MD: −1.79) in patients with hypertension when compared to usual care. It was not possible to ascertain an association of NT with change in SBP. 

The effectiveness of NT in the management of T2DM and hypertension have been investigated in systematic reviews, although none had focused exclusively on the primary care setting. Patients with T2DM and hypertension are mostly managed in the primary healthcare level, and access to specialized medical nutritional therapy is limited [18]. The lack of training on nutritional advice and patient overload often makes lifestyle interventions challenging for primary care nurses and physicians [11].

The systematic review by Kalyoncu et al. investigated the effectiveness of nutrition-based practices in prevention of hypertension but focused on healthy young adults [21]. Nicolson et al. performed a meta-analysis of five studies comparing antihypertensive medication versus lifestyle interventions (without antihypertensive), and likewise had inconclusive results on SBP reduction [22]. However, distinct from the review form Nicolson et al., in our review NT was often offered in combination to antihypertensive medications as part of the standard care at the primary care clinic. Besides, our review included more studies published after 2004.

Previous systematic reviews investigating the effectiveness of NT in T2DM had focused mainly on weight loss among obese patients [17], or T2DM prevention among healthy patients at risk for T2DM [19]. In addition, the benefit of group-based self-management education in reducing HbA1c had been previously indicated in a systematic review [20]. However, the inclusion criteria of this previous systematic review were restricted to group-based educational programs [20]. Moreover, two robust studies were published after this systematic review was carried out, which were now included in our analyses [34,47]. Furthermore, García-Molina et al. conducted a meta-analysis of studies assessing the effectiveness of lifestyle intervention (educational programs) in T2DM management [18]. García-Molina et al. found a weighted mean difference of −0.51 in HbA1c, a reduction that is more pronounced than the results from our meta-analysis. Although the inclusion criteria were similar to those used in our review, our review excluded studies conducted in settings other than the primary care, assessed separately patients with hypertension and applied the GRADE methodology to assess the quality of the evidence. 

Among the studies included in our meta-analyses, three studies had clustered samples (i.e., subjects were randomized at a group level), but the results were analyzed at an individual level [32,43,50]. By not taking clustering into account, the results may show significance where none exists [70]. Moreover, in one study there was no allocation concealment (i.e., subjects with even numbers were allocated to the intervention arm, while odd number subjects to the control arm) [45]. The individual-level analysis in a clustered sample in Hörnsten et al. [43], and the lack of allocation concealment in Javaid et al. [45], may explain the overestimated effect on HbA1c in both studies identified in the meta-analysis. Hence, we conducted sensitivity analysis, and the effect in favor of the intervention was maintained. 

Some limitations to our results must be acknowledged. First, we could not evaluate the effect of NT on hard outcomes such as mortality and frequency of cardiovascular events. The surrogate continuous outcomes evaluated allow a limited interpretation of the real clinical impact of the interventions. For HbA1c, a change of 0.5% is considered as clinically significant [71,72]. For blood pressure, a reduction of 10 mmHg is associated with a reduction of 17% in coronary artery disease incidence, 27% in stroke, 28% in heart failure and 13% in all-cause mortality [73]. When considering these cutoffs to extrapolate the correlation of blood pressure and HbA1c reduction on long term complications and mortality, the effect of NT is arguable. For instance, the mean change in DBP in patients with hypertension identified in our meta-analysis, although statistically significant, was notably lower than what could be regarded as clinically significant. Second, during data extraction, it was not possible to ensure that the studies considered equivalent variables for the calculation of the MAD.

Third, there was noticeable variability among the educational programs provided by the included studies, regarding intensity of the intervention, the professional responsible for delivering the intervention and the content of the intervention. While some educational interventions followed a nutrition education curriculum [48,52], others were based on motivational interviewing [42], or culturally tailored self-management interventions [35,53,58]. Moreover, some programs provided broad lifestyle modification advice, entailing promotion of regular physical activity. The choice of the best suited intervention might be influenced by previous local practices, logistic capacities, and cultural aspects, although the common aspect of these educational and counseling programs was that the NT was offered as part of a coordinated care in a primary setting [11]. 

Lifestyle modification is a central aspect to the management of patients with T2DM and hypertension [12]. Although the implementation of lifestyle and nutritional interventions in the primary care setting is challenging, the effect of educational programs on the management of hypertension (i.e., reduction in DBP) and T2DM, and food replacement for the management of T2DM have been extensively studied, although future studies could help establish the effect of NT on SBP. These results may contribute to further implementing NT in the primary care setting. Moreover, NT programs tailored for the primary care should be encouraged.

## 5. Conclusions

Nutritional therapy strategies (i.e., educational/counseling programs and food replacement programs) in primary care improved the glycemic control in patients with T2DM and educational programs in primary care improved DBP in individuals with hypertension. NT programs tailored for the primary care should be encouraged.

## Figures and Tables

**Figure 1 ijerph-19-04243-f001:**
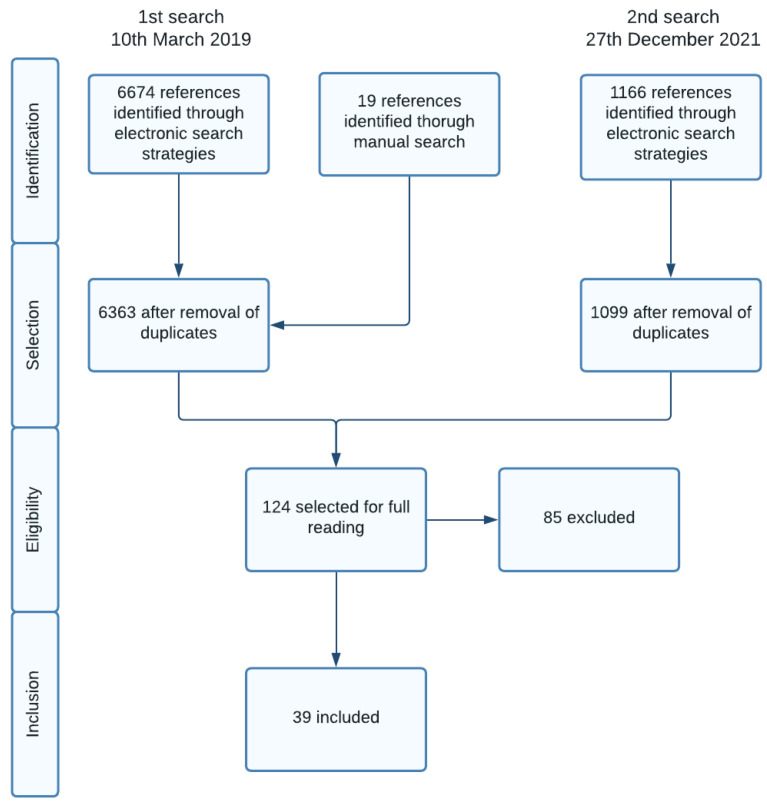
Study selection process.

**Figure 2 ijerph-19-04243-f002:**
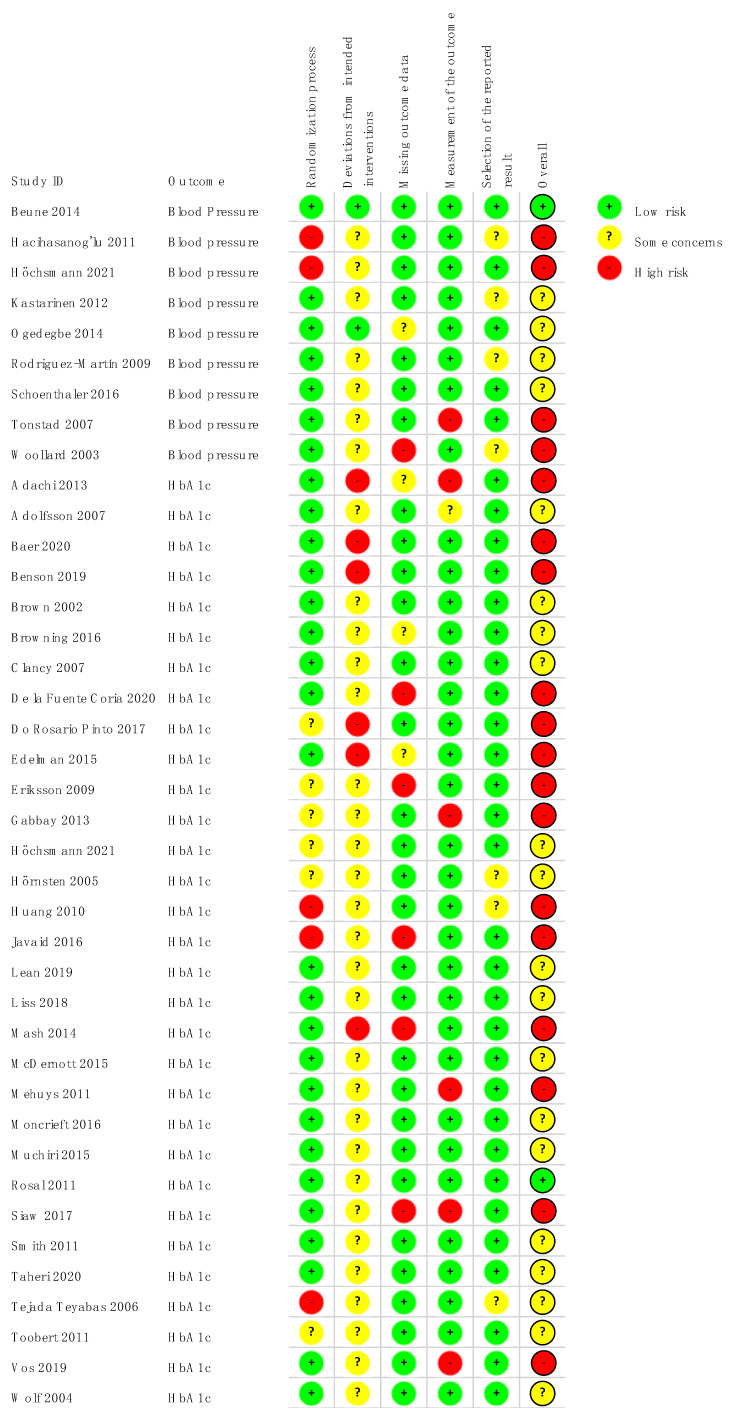
Risk of bias of included studies according to the main outcomes analyzed.

**Figure 3 ijerph-19-04243-f003:**
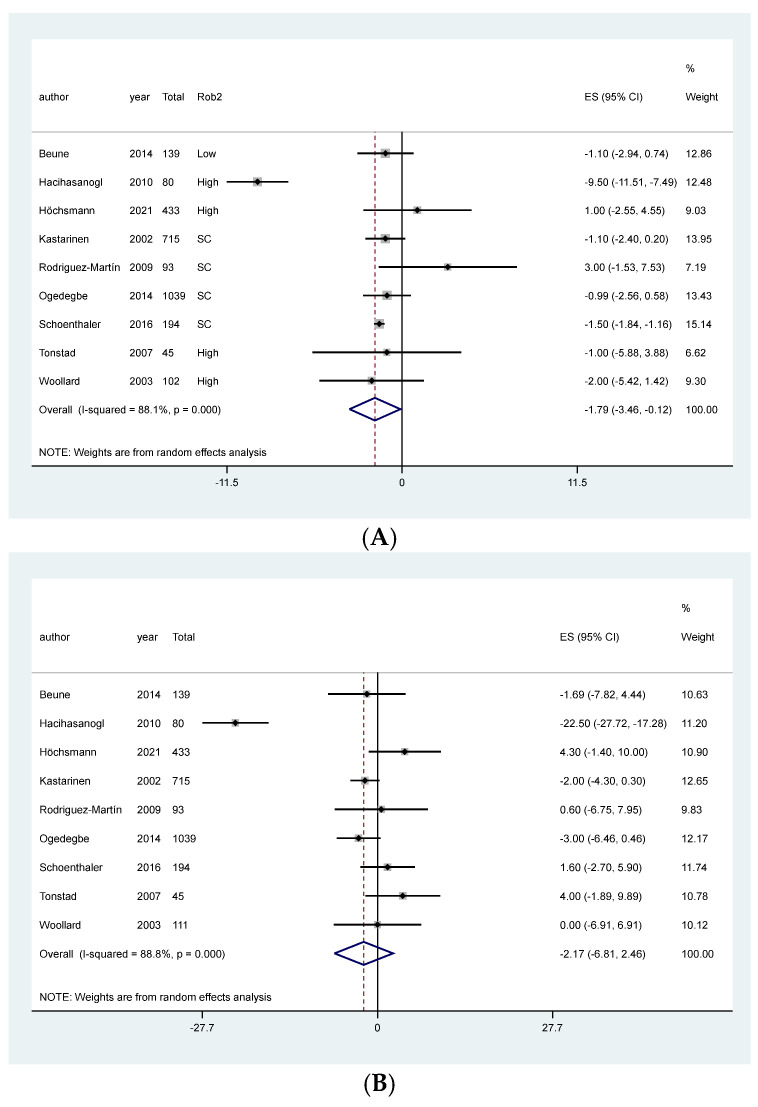
Meta-analysis of the effect of NT on blood pressure in patients primarily diagnosed with hypertension. (**A**) Diastolic blood pressure, (**B**) Systolic blood pressure.

**Figure 4 ijerph-19-04243-f004:**
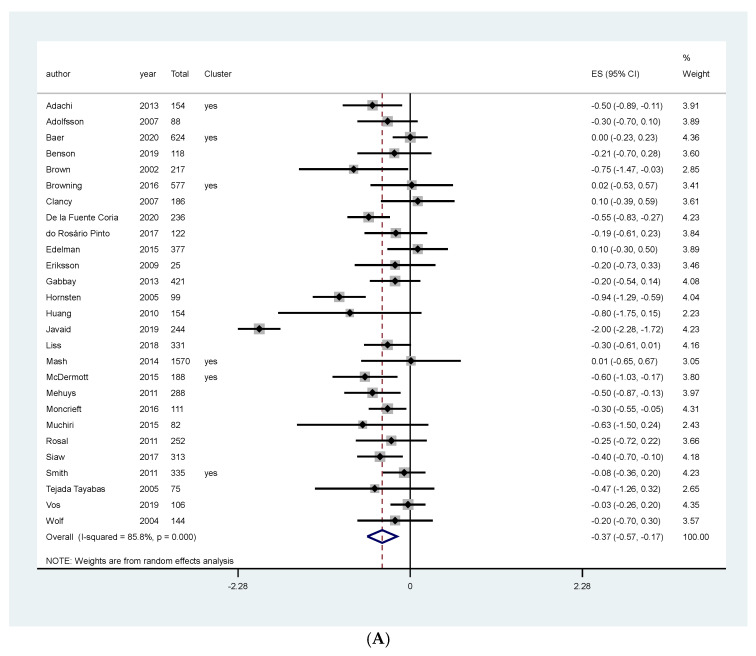
Meta-analysis of the effect of NT on HbA1C in patients primarily diagnosed with type 2 Diabetes Mellitus according to the type of intervention. (**A**) Counseling and educational programs, (**B**) Food replacement.

**Table 1 ijerph-19-04243-t001:** Summary of findings and quality of the evidence.

Nutrition Therapy Compared to Usual Care for Type 2 Diabetes and/or Hypertension						
**Patient or population:** Patients with type 2 diabetes and/or hypertension. **Setting:** Primary care **Intervention:** Nutrition therapy (counseling/educational programs in type 2 diabetes; food subistitution). **Comparison:** Usual care						
Outcomes	Anticipated Absolute Effects * (95% CI)		Relative Effect (95% CI)	№ of Participants (Studies)	Certainty of the Evidence (GRADE)	Comments
	Risk with Usual Care	Risk with Nutrition Therapy				
HbA1c (%) in counseling/educational programs follow-up: mean 14 months		MD **0.37 lower** (0.57 lower to 0.17 lower)	-	7437 (27 RCTs)	⨁⨁⨁◯ Moderate ^a^^,^^b^	In primary care, counseling/educational programs likely result in a reduction in HbA1c (%) in type 2 diabetes patients.
HbA1c (%) in food replacementfollow-up: mean 18 months		MD **0.54 lower** (0.75 lower to 0.32 lower)	-	440 (2 RCTs)	⨁⨁⨁◯ Moderate ^c^	In primary care, food replacement likely results in reduction in HbA1c (%) in type 2 diabetes patients.
Systolic blood pression in participants with hypertensionfollow-up: mean 14 months		MD **2.17 lower** (6.81 lower to 2.46 higher)	-	4518 (9 RCTs)	⨁⨁◯◯ Low ^d^^,^^e^	
Diastolic blood pression in participants with hypertensionfollow-up: mean 14 months		MD **1.79 lower** (3.46 lower to 0.12 lower)	-	2840 (9 RCTs)	⨁⨁⨁◯ Moderate ^d^^,^^f^	In primary care, counseling/educational programs likely reduce slightly diastolic blood pression in participants with hypertension.
**GRADE Working Group grades of evidence** **High certainty:** we are very confident that the true effect lies close to that of the estimate of the effect. **Moderate certainty:** we are moderately confident in the effect estimate: the true effect is likely to be close to the estimate of the effect, but there is a possibility that it is substantially different. **Low certainty:** our confidence in the effect estimate is limited: the true effect may be substantially different from the estimate of the effect. **Very low certainty:** we have very little confidence in the effect estimate: the true effect is likely to be substantially different from the estimate of effect.						
**Explanations** All included studies were classified as having either some concerns or high risk of bias according to RoB2. Although the meta-analysis showed a high statistic heterogeneity (I-squared = 85.8%), it was attributed to two studies that overestimated the intervention effect. The meta-analysis without these two studies maintained the effect in favor of the intervention (Appendix A). As a result of this the quality of evidence was not rate down due to inconsistency. The two included studies were classified as having some concerns according to RoB2 Most included studies were classified as having high risk of bias according to RoB2. The 95% CI overlaps no effect, but it fails to exclude important benefit. The summary effect crossed the line of the null effect. Although the meta-analysis showed a high statistic heterogeneity (I-squared = 88.8%), it was attributed to a study that overestimated the intervention effect. The meta-analysis without this study maintained the null effect (Appendix A). As a result of this the quality of evidence was not rate down due to inconsistency. Although the meta-analysis showed a high statistic heterogeneity (I-squared = 88.1%), it was attributed to a study that overestimated the intervention effect. The meta-analysis without this study maintained the effect in favor of the intervention (Appendix A). As a result ofof this the quality of evidence was not rate down due to inconsistency.						

* The risk in the intervention group (and its 95% confidence interval) is based on the assumed risk in the comparison group and the relative effect of the intervention (and its 95% CI). Abbreviations: CI: confidence interval; MD: mean difference.

## Data Availability

All the data generated during this study is provided in the main manuscript and its Appendix A.

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
