# Peer review of "Effectiveness of Strategies for Nutritional Therapy for Patients with Type 2 Diabetes and/or Hypertension in Primary Care: A Systematic Review and Meta-Analysis"

_ijerph, 2022, doi:10.3390/ijerph19074243_

Round 1

Reviewer 1 Report

I am concerned that the importance of T2 DM or any health condition is quoted in isolation from overall health reporting. A quick check with the WHO Global Diabetes report shows the same approach.

T@DM is undoubtedly a large and growing international problem, but it is vital to keep reports in perspective

The issue arises again with the report of direct health expenditure on T2DM. There is no total health expenditure to place this large number in context. The source, reference (5) does likewise.

Author Response

Dear reviewer,

Thank you for reviewing our manuscript. The reviewer is right, and the introduction was amended accordingly (cost of diabetes is reported in relation to GDP) (lines 41-42). We stay at your disposal for further clarifications.

Reviewer 2 Report

Dear Authors:

Regarding the manuscript with the title “Effectiveness of strategies for nutritional therapy for patients with type 2 diabetes and/or hypertension in primary care: a systematic review and meta-analysis”, I have 2 major concerns and some minor comments to address.

Major Comments:

Comment 1:

Line 69 to 71: As the paper referred to a systematic review, the following sentence must be deleted “Although several randomized trials have evaluated the effectiveness of NT in the management of DM and hypertension [17-20], the effectiveness of NT in the primary care setting was yet to be investigated in a systematic review”.

Instead of focusing on randomized controlled trials authors must emphasize the existence of several systematic reviews regarding nutritional interventions on type 2 diabetes or hypertension patients. To clarify the pertinence of this paper, authors must emphasize what this paper adds to the previous systematic reviews that exist regarding this theme (In what way does this paper differ from the others?). This is crucial for the pertinence of this paper. 

Examples of Systematic reviews on hypertension:

“Lifestyle interventions or drugs for patients with essential hypertension: a systematic review”

" A systematic review of nutrition-based practices in prevention of hypertension  among healthy youth"

Examples of Systematic reviews on type 2 diabetes:

“Lifestyle weight-loss intervention outcomes in overweight and obese adults with type 2 diabetes: a systematic review and meta-analysis of randomized clinical trials”

“Lifestyle interventions for patients with and at risk for type 2 diabetes: a systematic review and meta-analysis”

“Improving type 2 diabetes mellitus glycaemic control through lifestyle modification implementing diet intervention: a systematic review and meta-analysis”

Comment 2:

Line 355-356: “This systematic review is to the best of our knowledge the first systematic review to evaluate the benefit of NT for patients with hypertension. Differently, the benefit of group-based self-management education in reducing HbA1c had been previously indicated in a systematic review [152].

Considering my major comment 1, authors must rephrase the previous sentence accordingly.

Minor Comments:

Comment 1:

On lines 19-20: On Abstract, authors referred 27 studies plus 2 studies, respectivelly. On line 17, authors stated the existence of 30 studies

Comment 2:

Line 22: Authors must change “ due to the risk of bias and imprecision, effect in SBP was unclear” by “The effect on SBP was unclear due to risk of bias and imprecision “

Comment 3:

Line 60: Authors must withdraw the word “prevention”, as the sample of this manuscript are patients with type 2 diabetes and hypertension still diagnosed

Comment 4:

Line 125: Authors must check the expression “on all 111 databases”. To which databases authors referred?

Comment 5:

Line 126: Regarding synonims, authors must give exemples. Besides, authors must add the words AND or OR between the descriptors used on the search.

Comment 6:

Line 127-128: In PubMed and Embase a filter for randomized controlled trials were applied. Why only on Pubmed and Embase this filter was applied?

Comment 7:

Authors must change “The following databases were searched for eligible studies” by “The following databases will also be searched for eligible studies”

Comment 8:

Regarding Figure 1, I have two questions:

What is the difference between the first and second search? These two search were not made according to the same descriptors

Why 2 years and 8 months of difference between the first and second search

Comment 9:

Line 204: Authors have not to indicate on References the studies that were excluded. Thus authors must change The 85 references excluded  by “The 85 references excluded”. Besides, all references must be deleted from the chapter of References. The number of all subsequent references must be changed accordingly.

Comment 10:

Line 204 and 205: Authors must change “the main reasons” by “the reasons”

Comment 11:

Lines 214-216: Authors named thirty nine studies, but referred eight studies for hypertension and 30 for type 2 diabetes!! As on line 218, authors referred nine studies for hypertension, including study with reference 145, I think this may be the error. I please ask authors to confirm this situation.

Comment 12:

Line 218: Authors used for the first time the acronym HTN. Authors have two options: use always the word hypertension OR the first time authors used the word hypertension on the manuscript (excluding Abstract), authors have to write hypertension (HTN)

Comment 13:

Line 221. Authors must indicate the studies that were conducted in groups and individually and by practice nurses, dieticians, or health educators.

Comment 14:

The focus of this paper is to find papers realized in primary care setting. I have one doubt regarding this aspect: Authors only included studies directly conducted on primary care settings or also included studies in which patients were recruited on primary care setting but the intervention was not conducted on primary care setting?

Reviewer 3 Report

The authors present a manuscript correlating nutritional intervention with the adverse effects of type II diabetes

However, there are various conditions that should at least be reviewed, as presented in this way is not of high significance

- The selection criteria of the studies are not clear, they should be better characterized on the single study

- Too different nutritional interventions are grouped together, for example saying vegetarian or vegan does not mean much, it should be emphasized if it is low-calorie or if it has a valid supply of fiber (both very important factors for the management of type II diabetes)

- The activity is not considered, a fundamental point both for the establishment and for the management of the pathology

- The conclusions would seem to provide a negligible contribution of a nutritional intervention on diabetes when instead it should be the basis of treatment with or without the use of drugs in combination

Round 2

Reviewer 2 Report

Dear Authors.

Thank you for the changes made in response to my comments. The manuscript was significantly improved. Regarding the actual form of the manuscript, I only have 2 minor comments to address.

Comment 1:

Lines 66-74: Regarding what is stated and very well on Discussion, I suggest that authors change “Systematic reviews have demonstrated the effectiveness of NT and lifestyle education on the management of T2DM [17-20] and hypertension[21,22], but none focused on the primary care setting. Patients with T2DM and hypertension are mostly managed in the primary health care level, and access to specialized medical nutritional therapy is limited [18]. The lack of training on nutritional advice and patient overload often makes lifestyle interventions challenging for primary care nurses and physicians [11]. Hence, the objective of this systematic review was to evaluate the effectiveness of NT programs delivered in the primary care setting in the management of patients with T2DM and/or hypertension.” by “Previous systematic reviews have demonstrated the effectiveness of NT and lifestyle education on the management of T2DM [17-20] and hypertension[21,22]. However, none focused exclusivelly on the primary care setting; some were conducted on specific populations, such as young adults [21], obese patients [17], patients at risk of type 2 diabetes [19]; inclusion criteria were restricted to group-based educational programs [20] and antyhipertensive medications take not part of the standard care [22]. Hence, the objective of this systematic review was to evaluate the effectiveness of NT programs delivered exclusivelly in the primary care setting in the management of patients aged ≥18 years old with T2DM and/or hypertension.

Comment 2:

Line 358: Authors must change “although none has focused on the primary health care setting” by “although none has focused exclusivelly on the primary health care setting.

Author Response

Dear reviewer, thank you for the constructive comments and the careful review. 

The two minor alterations were carried out as suggested (lines 66-71 and line 358). 

Reviewer 3 Report

I think that the authors made enough improvement, but some problems still remain.....

Author Response

Dear reviewer, thank you for the careful review and your comments.